# Multimodal Management of Fecal Incontinence Focused on Sphincteroplasty: Long-Term Outcomes from a Single Center Case Series

**DOI:** 10.3390/jcm11133755

**Published:** 2022-06-28

**Authors:** Carlos Cerdán Santacruz, Débora M. Cerdán Santacruz, Lucía Milla Collado, Antonio Ruiz de León, Javier Cerdán Miguel

**Affiliations:** 1Colorectal Surgery Department, Clínica Santa Elena, 28003 Madrid, Spain; fjcerdan@hotmail.com; 2Colorectal Surgery Department, Hospital Universitario de la Princesa, C/Diego de León 62, 28006 Madrid, Spain; 3Neurology Department, Hospital General de Segovia, 47002 Segovia, Spain; deboracerdan@hotmail.com; 4Thoracic Surgery Department, Hospital Central de la Defensa, 28047 Madrid, Spain; luciamillacollado@hotmail.com; 5Gastroenterology Department, Hospital Clínico Universitario San Carlos, 28040 Madrid, Spain; aruizdeleon@gmail.com

**Keywords:** anal incontinence, sphincteroplasty, CCIS, biofeedback, sacral nerve stimulation, posterior tibial nerve stimulation

## Abstract

The management of patients with fecal incontinence and an external anal sphincter (EAS) defect remains controversial. A retrospective series of overlapping anal sphincteroplasties performed between 1985–2013 from a single center, supplemented by selective puborectalis plication and internal anal sphincter repair is presented. Patients were clinically followed along with anorectal manometry, continence scoring (Cleveland Clinic Incontinence Score—CCS) and patient satisfaction scales. Patients with a suboptimal outcome were managed with combinations of biofeedback therapy (BFT), peripheral tibial nerve stimulation (PTNS), sacral nerve stimulation (SNS) or repeat sphincteroplasty. There were 120 anterior sphincter repairs with 90 (75%) levatorplasties and 84 (70%) IAS repairs. Over a median follow-up of 120 months (IQR 60–173.7 months) there were significant improvements in the recorded CCIS values (90.8% with a preoperative CCIS > 15 vs. 2.5% postoperatively; *p* < 0.001). There were 42 patients who required ancillary treatment with four repeat sphincteroplasties, 35 patients undergoing biofeedback therapy, 10 patients treated with PTNS and three managed with SNS implants with an ultimate good functional outcome in 92.9% of cases. No difference was noted in ultimate functional outcome between those treated with sphincteroplasty alone compared with those who needed ancillary treatments (97.1% vs. 85.7%, respectively). Overall, 93.3% considered the outcome as either good or excellent. Long-term functional outcomes of an overlapping sphincteroplasty are good. If the initial outcome is suboptimal, response to ancillary treatments remains good and patients are not compromised by a first-up uncomplicated sphincter repair.

## 1. Introduction

First described by Parks and McPartlin [1], the traditional surgical management in patients presenting with fecal incontinence (FI) where there is an external anal sphincter (EAS) defect is an overlapping sphincteroplasty [2]. Although short- and medium-term functional outcomes after surgery are acceptable, some reports have shown that patients can anticipate a significant fall-off in the success of sphincter repair over a more prolonged follow-up [3,4,5]. Given the multifactorial nature of FI, the repair of a demonstrable sphincter defect remains valid [6]. There is considerable evidence that the success of sphincteroplasty, however, is dependent upon the level of experience of the surgeon [7] and the grade of the sphincter tear [8]. Nevertheless, although results with this technique appeared to deteriorate over time, the percentage of patients who remained satisfied was high [5].

By comparison, sacral neuromodulation (SNM) was initially introduced into FI management for those patients with intact anal sphincters [9] but over time there has been an extension of its indications as a primary therapeutic alternative to include patients with an EAS defect [10,11,12]. In the management of such a patient, however, there have been those who have argued with equal conviction both the pros and cons of SNM as a first-up treatment over delayed sphincter repair [6,13]. For a benign condition such as FI, the subjective satisfaction reported by the patient remains an important indicator where it is accepted that the standards judged for success between sphincteroplasty and SNM are not always strictly comparable [14] and where few treatments for incontinence will achieve a perfect result. Patients should also be advised regarding the known outcomes of salvage procedures when initial conservative therapy or individual surgical treatments have failed. This study analyzes the functional outcome of anal sphincteroplasty in a closely followed patient cohort managed by a single surgeon over a prolonged period with assessment of the clinical value of secondary complementary therapies if and when they were needed.

## 2. Materials and Methods

Ethical permission was obtained from the local hospital Ethics Committee for the conduct of this retrospective analysis, and it was registered in Clinicaltrials.gov (NCT04727463). Data were retrieved from a prospectively maintained database incorporating an unselected group of consecutive cases presenting with a history of severe fecal incontinence (FI) where there was an associated external anal sphincter (EAS) defect and who underwent an overlapping sphincteroplasty as definitive incontinence management. The study included all patients derived from a tertiary colorectal referral practice in a 900 bed University-affiliated hospital who were registered between January 1985 and December 2013 with all cases managed by a single surgeon (JCM) over this time period. The results are reported in accordance with the Strengthening the Reporting of Observational Studies in Epidemiology statement for observational studies [15].

Cases were included after a failed period of conservative management which included dietary changes, constipating medication and biofeedback therapy (BFT). All consecutive patients operated on for fecal incontinence who underwent a sphincter repair were included.

Demographic data were collated (age, gender) along with the duration of symptoms, prior gynecologic and/or proctologic surgery, previous obstetric history and the status of sexual activity. The severity of incontinence was assessed by the Cleveland Clinic Incontinence Score—CCIS [16] separating the score into three broad range categories (0–8; 9–15 and 16–20) similar to Rothbarth et al. [17]. The Visual Analogue Scale was also recorded at baseline. Sphincter morphology was evaluated by digital rectal examination with careful palpation of the external anal sphincter (EAS) in each case. Endoanal ultrasound (EUS) has been available since the beginning of 1997 and was performed in 81 cases. All patients underwent preoperative anorectal manometry, and resting pressure (RP), maximum squeeze pressure (MSP) and sphincter high-pressure zone length (SL) were determined.

### 2.1. Surgical Technique

The policy involved the performance of an overlapping EAS sphincteroplasty with puborectalis plication along with selective internal anal sphincter (IAS) suture when injured. All patients underwent a mechanical bowel preparation with oral sodium phosphate along with the administration of perioperative intravenous antibiotic prophylaxis. Procedures were variably performed under regional or general anesthesia with patients catheterized and placed in the lithotomy position.

The perineum was infiltrated with a 1:200,000 adrenaline-saline solution with a transverse incision made in the region of the perineal body and with sharp dissection (and protection) of the posterior vaginal wall, separating it from the anterior anorectum. When there existed an important or pathologic scar, it was completely excised in order to get a free tension sphincter and to allow further healing over healthy tissue. The EAS was minimally mobilized in order to avoid denervation and/or devascularization, but enough to allow a tension-free sphincteroplasty with an overlapping two-layer sphincter suturing with 2/0 Vicryl sutures. During this part of the intervention, sequential checking of tension is mandatory in order to identify that critical moment of not extensive but enough dissection of both EAS limbs. The puborectalis muscle was approximated towards the midline with normally two additional sutures. Any IAS defect anteriorly was closed using 4/0 interrupted Vicryl. In some cases, the surgical wound was left either partially or completely open and allowed to heal by secondary intention. In those cases where there was destruction of the perineal body a Corman-style advancement perineoplasty was performed as a random pattern transposition flap [18]. This approach was used to reconstruct the perineal body and superficial perineal musculature as well as to restore the distal part of the rectovaginal septum by separating the anal and vaginal walls. Figure 1 shows the performance of the anterior sphincteroplasty and puborectalis plication. All procedures were performed without proximal diversion. The urinary catheter was left in situ for 48 h with introduction of a regular diet at that time and the discretionary use of laxatives in the immediate postoperative period.

Those who were sexually active were advised to avoid sexual intercourse for the first three postoperative months. Patients were followed up regularly in the clinic to observe wound healing with ultimately six monthly assessments for the first three years and then annual checks thereafter. At each visit patients were clinically examined with determination of the CCIS. Anorectal manometry was conducted annually for the first five years with anal endosonography performed one year after surgery (unless otherwise indicated). Sonography was repeated in those with deterioration in their continence function. In those cases where there was a mild continence deterioration (<4 CCIS points change) dietary means and constipating agents were used for control, whereas in patients with a worse decline in function (>5 CCIS points change), BFT was initially used. In those patients with disruption of the sphincter repair, a repeat sphincteroplasty was usually attempted after a 12-month period. In other circumstances and in the event that there was no improvement with BFT, either posterior tibial nerve stimulation (PTNS) or SNS was offered. At the conclusion of the study either face-to-face or telephone interviews were conducted to assess the final CCIS and to determine the Visual Analogue Scale (VAS). Patients were also asked if given the same clinical circumstances they would undergo the same procedure.

### 2.2. Statistical Analysis

The analysis of data was performed using the SPSS Version 20.0 for Windows software (Chicago, IL, USA). Qualitative variables are presented by frequency distribution. Quantitative variables are summarized by means (+SD) or medians (plus interquartile range—IQR). Comparative pre- and postoperative paired data were assessed by the Student’s *t*-test with ANOVA employed for repeated measures after Bonferroni correction. Preoperative manometric values were compared with the last available registered postoperative recordings. Categorical variables were compared with the χ^2^ or Fisher’s Exact test where appropriate, with *p* values < 0.05 considered significant.

## 3. Results

Over the period of analysis, 113 women (94.2%) and seven men (5.8%) underwent a total of 120 anal sphincter repairs (median overall age 59, range 16–84 years). Of the cohort, 56 (46.6%) were >60 years of age and 17 (14.2%) >70 years of age at the time of surgery. In presentation, 97.5% complained of incontinence to solid stool, 99.2% to liquid stool and 100% had gaseous incontinence, with 88% reporting urgency of defecation and 65% reporting regular soiling. The median duration of incontinence symptoms was 120 months (range 10–480 months). The cause of the sphincter injury was obstetric-related in 98 (81.6%) cases, following anal fistula surgery in 10 (8.3%), after other proctologic surgery (abscess drainage, internal anal sphincterotomy and hemorrhoidectomy) in eight cases (6.6%), and with perineal trauma in four (3.3%) cases. Table 1 shows the clinical features and demographics of the operated cohort outlining their preoperative CCIS and the presence of concomitant pelvic floor disorders on clinical examination, as well as the surgical procedures performed to improve continence as well as to deal with ancillary anorectal pathology.

The EAS defect was anteriorly located in 105 (87.5%) cases, lateral in 11 (9.2%) and posterior in for (3.3%). The mean angle of separation of the muscle ends was 126° (range 90–180°; SD = 28°). An overlapping sphincteroplasty (OS) was performed in 119 cases (99%), with one direct apposition repair and with a concomitant levatorplasty in 90 (75%) of the patients. An IAS repair was performed in 84 (70%) cases. There was one patient suffering from a hemorrhage requiring a return to the operating theatre with hemostasis based on suturing. There were 24 (20%) wound infections, all of which healed within two to six weeks by secondary intention. Of the cohort, 105 (87.5%) were available for follow-up, with eight patients who died from unrelated events including one with chronic dementia who was unavailable for assessment along with six patients lost to follow-up. The median follow-up period was 120 months (IQR 60–173.7 months), with the shortest follow-up of 24 months and the longest at 372 months. The follow-up was >five years in 95 (79%) and >10 years in 60 (50%) of the cases.

Table 2 shows the recorded CCIS values demonstrating significant improvement at all measured postoperative time periods when compared with the baseline (*p* < 0.001). This is shown graphically in Figure 2. The baseline and last follow-up CCIS values were compared in three symptomatic severity groups. There were 90.8% with a preoperative CCIS > 15 compared with 2.5% postoperatively (*p* < 0.001). Preoperatively, there were 9.2% with a recorded CCIS between 9–15 compared with 6% of postoperative cases (*p* < 0.001) and 0% of cases with a preoperative CCIS < 8 compared with 91.5% of postoperative cases (*p* < 0.001).

In the assessment at five years, there was a significant improvement in the CCIS when compared with baseline for patients undergoing all types of repairs (overlapping sphincteroplasty, *p* = 0.05; levatorplasty, *p* = 0.006 and IAS plication, *p* = 0.009). Of those undergoing a sphincteroplasty alone, 78 (65%) reported good continence, with the remaining 42 requiring ancillary treatment (Figure 3). In this group, four patients underwent a repeat sphincteroplasty with a mean period between surgeries of 109 months (range 54–132 months) and with good functional outcome in two cases. Overall, an acceptable functional outcome was achieved in 92.9% of cases at final evaluation using a variable combination and sequence of BFT, SNS and PTNS therapies, with eight patients with an unsatisfactory outcome. There was no difference in the likelihood of a successful functional outcome between those undergoing a sphincteroplasty alone and those who required ancillary treatments (97.1% vs. 85.7%, respectively). No significant differences were noted in the resting or squeeze manometric evaluations when compared with baseline, however, there was a significant increase in the mean HPZ length from 2.6–3.2 cm (*p* = 0.032).

Table 3 shows the results of a univariate analysis to determine which factors correlated with either the need for an ancillary treatment over and above surgery or with a good outcome adjudged as a VAS score of seven or more. There were no specific clinical or operative factors identified for either category.

Visual analogue scale (VAS) scores changed from a median value of 1.55 and a 100% scoring below 4 in the preoperative assessment, to a median value of 8.73 and 89.5% of the sample scoring 8–10 in the last follow-up rating (*p* < 0.001). In all, 104/105 patients questioned would undergo the same procedure again under similar circumstances.

## 4. Discussion

The functional results of overlapping external anal sphincteroplasty over a 35-year period for an unselected group of patients presenting with fecal incontinence (FI) are presented. Overall, there was an acceptable functional outcome in 92.2% of the 120 repairs with a median follow-up period of 120 months where 80% of cases were assessed for longer than five years. In the cohort, 35% of cases required ancillary postsurgical treatment for incontinence which variably included biofeedback therapy (BFT) and selective peripheral tibial nerve stimulation (PTNS) and sacral neuromodulation (SNS). Of this group overall there was a good reported functional outcome in 85.7% of cases. To our group, these results are perfectly consistent with the multifactorial origin of fecal incontinence [19]. Nevertheless, in the presence of sphincter damage, we are convinced that a previous sphincteroplasty would contribute positively to the overall outcome [19].

In the event of anal incontinence, it is necessary to carry out an individualized assessment. If there is a sphincteric injury and conservative measures have failed, surgical treatment is indicated in our group, following the previously mentioned systematics. However, some exceptions can also be considered. As absolute contraindication a separation of the sphincteric margins >180° or multiple sphincteric injuries can be mentioned. Regarding relative contraindications, it would be necessary to determine them personally in each patient, but severe defecatory dyssynergia or decreased rectal sensation verified by anorectal manometry can be prioritized for alternative treatments better than sphincteroplasty as first line. Nevertheless, in general terms, if there is a sphincteric injury, our opinion is that it is necessary to carry out the most complete and satisfactory anatomical repair possible and, based on the results, establish complementary treatments as necessary, as has been previously elucidated in our work.

Regarding the time to perform the sphincteroplasty, it is obvious that live recognition of the sphincteric injury, whether due to obstetric trauma, accident of any kind or violent insult, it should be repaired immediately, although there do not exist comparative data to answer this issue; whether an immediate or a delayed repair could result in a better outcome has not previously been analyzed, and therefore any judgement in this regard would be speculation. Nevertheless, it is quite common that most patients with anal incontinence discovered later; in our series, almost half of the patients were older than 60 years, with a median duration of symptoms of 120 months. Therefore, sphincter repair should be performed based on incontinence severity, with no differences found regarding the time elapsed since the injury or patients’ age.

A delayed anterior sphincter repair either alone or combined with puborectalis plication and with individualized IAS repair still remains a good option for patients presenting with FI and an obstetric-related perineal injury.

Both plication of IAS and levatorplasty are exceptional in the literature. IAS plication is usually performed on an individual basis [20,21,22,23], and levatorplasty is mostly employed in older females to avoid dispareunia, although specific indications or selection criteria have not been established [20,21,22,24,25,26,27,28]. However, Evans et al. [29], Miller et al. [30] and more recently Berg et al. [14] systematically performed this intervention in their series, obtaining satisfactory results reflected both in the increasing sphincter pressures and, especially in the case of Evans, the long-term maintenance of the results.

It has been the authors’ practice to use absorbable sutures, as we have always considered them to produce less local tissue inflammation and granuloma, which might worsen healing and, eventually, the final result. Apart from this consideration, strained or monofilament choice should not determine the final result.

The long-term global results obtained in our series are considerably higher than those previously reported. Although good results of around 75–96% have been reported in the immediate postoperative period, these have tended to subsequently decrease with values commonly reported of around 35–50% [3,4,14,21,22,24,25,26,28,30,31,32,33,34,35].

Despite the fact that postoperative results might decline over time, Oom et al. [22] demonstrated a high satisfaction rate in a medium-term follow-up in 172 cases where three-quarters were available for assessment. This effect was noted because of a marked reduction in the frequency of incontinence episodes.

In the modern era, SNS is seen as a competitive alternative in those FI cases with a demonstrable EAS defect [36], but it should be remembered that broad use of neurostimulation may report suboptimal results [37] where there is a moderately high incidence of an initial lack of efficacy, or even an ultimate loss of efficacy in some cases. In the study reported by Maeda et al. [37] of a cohort of patients undergoing SNS, the combined number of those with a suboptimal result and an adverse event remained high even though the ultimate functional outcome was good or acceptable. This study also reported an ultimate success rate of only 34.1% when secondary interventions became necessary.

Our results are also consistent with others demonstrating in successful cases a non-definitive improvement after surgery in measurable manometric parameters [38]. In these patients the effects of surgery may be to increase the functional length of the anal canal. The use of sphincter repair in experienced hands would appear to be a valid, low cost management option with durable long-term success [4,14,21,25,33,39]. Our evidence would also support the fact that a sphincteroplasty does not clinically compromise the patient further along and that ancillary treatments such as BFT can then be effective, particularly as patients adapt to the newly configured anatomy of an overlap repair [21,33].

Another important aspect is the option of simultaneously treating any other anorectal and/or perineal disorders. This is something rarely cited [40], and is impossible when opting for other treatments for FI such as SNS. In effect, in 33 cases (27.5%), an additional surgical procedure was performed in our series without any impact on the sphincteroplasty or its clinical outcomes.

It should be noted that the procedure generally evolves with minimal complications, most of which are easily resolvable, including wound infection in around 20% of cases, a value similar to others reported [3,20,21,24,41].

The exhaustive postoperative follow-up is another detail which we consider key to obtaining the best results after a sphincteroplasty, something that has not been systematically reported in the literature. We maintained a strict and prolonged follow-up of our patients which ranged from 24–372 months, with an average of 128 months. This lengthy control enabled us to immediately and objectively identify any type of deterioration and to implement measures to resolve or alleviate any problems that arose, including hygiene, dietary and medication recommendations, BFB, neuromodulation or a second sphincteroplasty. This has been exceptionally considered in literature [23].

Despite the value of a prolonged follow-up by a dedicated team, it is accepted that our study is limited by its small numbers, although it is one of the largest published ones, and also by its retrospective design. It would have been interesting to present the results of QoL or patient-reported outcome measures [42], but as it is a so long-time Cohort, none of these tools existed and therefore we substituted this with the VAS.

## 5. Conclusions

The long-term outcome of anatomical surgical repair of the anal canal in expert hands, based on overlapping sphincteroplasty plus puborectalis plication and individualized IAS repair is good. As a functional multifactorial affliction, patients should be counseled about the fact that continence following a sphincter repair might decline over time, but they have to be confident that an uncomplicated sphincteroplasty does not compromise their eventual functional outcome. The parameters for judging functional success after sphincter surgery appear harsher than those adopted for SNS, ref. [9] but despite this caveat, if further ancillary treatments (BFT, PTNS, or SNS) are needed postoperatively, the ultimate outcome remains acceptable. Following these principles, excellent clinical results and patient satisfaction can be achievable.

## Figures and Tables

**Figure 1 jcm-11-03755-f001:**
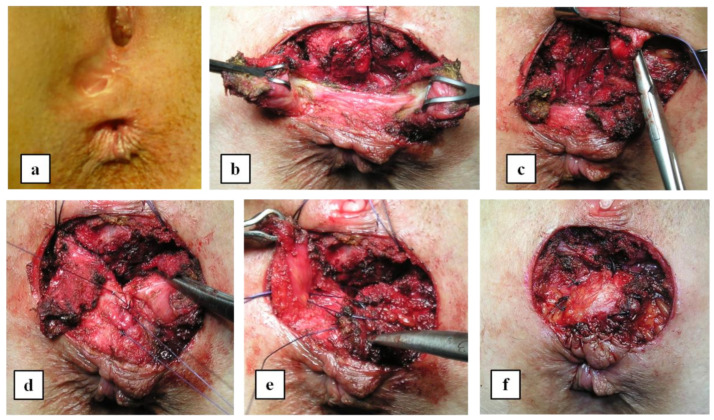
Detailed key technical points concerning an overlapping sphincteroplasty. (**a**): Basal anal inspection. Scar tissue secondary to episiotomy and “smoothing” of anal margin should be noted. (**b**): Dissection of both puborectalis limbs and the ends of the external anal sphincter. (**c**): Levatorplasty with interrupted sutures. (**d**): Internal anal sphincter plication. (**e**): Overlapping sphincteroplasty. (**f**): Final result. The wound is left totally or partially open. No drains are used.

**Figure 2 jcm-11-03755-f002:**
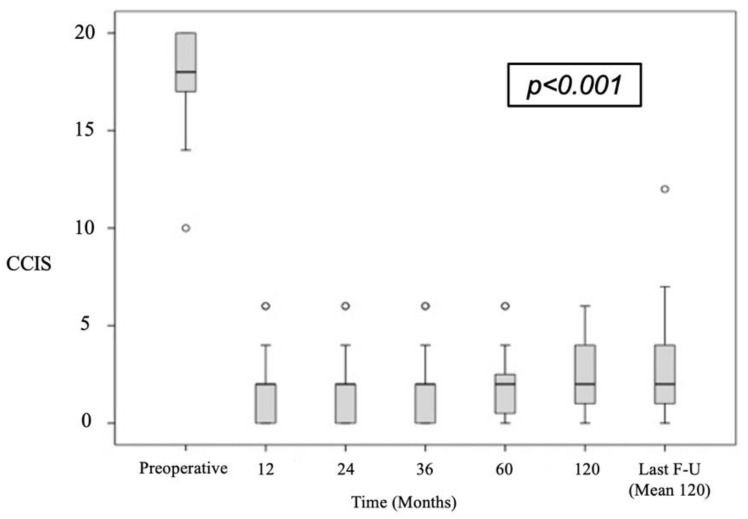
Mean values and 95%-CI of preoperative and each follow-up CCIS determination.

**Figure 3 jcm-11-03755-f003:**
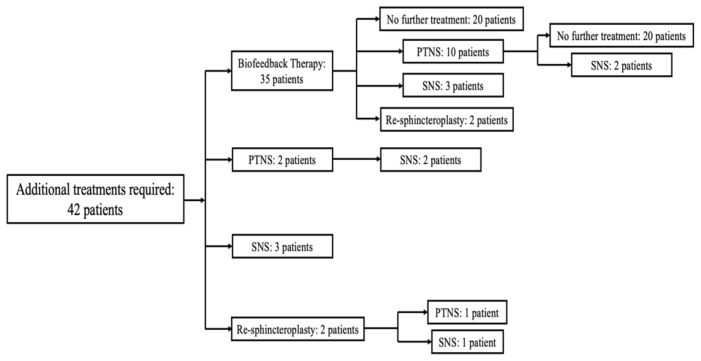
Flow-diagram of patients who needed additional treatments after initial sphincteroplasty. PTNS: Posterior tibial nerve stimulation; SNS: Sacral nerve stimulation.

**Table 1 jcm-11-03755-t001:** Demographic, preoperative and operative data for the whole group.

			*n* = 120
**Age (Years) ***			59 (16–84)
Gender		Male	7 (6)
		Female	113 (94)
Duration of symptoms before surgery (Months) *			120 (10–480)
Baseline CCIS		Total *	18 (9–20)
		0–8	0 (0)
		9–15	11 (9.2)
		>15	109 (90.8)
Preoperatory Manometric Variables		RP (LNR: 65 mmHg) ^‡^	27 (20–35)
		MSP (LNR: 140 mmHg) ^‡^	50 (38–64)
		SL (cm) ^‡^	2.6 (2–3)
Endoanal Ultrasound (81)		EAS defect	81 (100)
		IAS defect	41 (50.6)
Previous Obstetric History ^¥^		Vaginal Deliveries	98 (86.7)
		Episiotomy (Range 1–5)	78 (69)
		Third-Fourth degree tear	41 (36.3)
Concomitant Pelvic Floor Disorders ^¥^		Rectocele	15 (13)
		Recto-Vaginal Fistula	10 (9)
		Pelvic floor descent	9 (8)
		Rectal Prolapse	4 (3.5)
		Enterocele	3 (2.6)
Anal Incontinence Surgical Techniques	Apposition Sphincteroplasty		1 (0.83)
	Overlapping Sphincteroplasty (119 cases/99.1%)	OSph Alone	12 (10)
		OSph + ALev	14 (11.6)
		OSph + IAS repair	16 (13.3)
		OSph + ALev + IAS repair	68 (56.6)
		OSph + TPFR	8 (6,6)
		OSph + Postanal Repair	1 (0.83)
Associated Pathologies Surgical Treatment		Rectocele	13 (10.8)
		Corman’s graft	8 (6.6)
		Recto-Vaginal Fistula	6 (5)
		Enterocele	2 (1.66)
		Hemorrhoidectomy	2 (1.66)
		Rectal Prolapse (Delorme)	1 (0.83)
		Rectal Villous Adenoma Removal	1 (0.83)

Data are expressed as number of patients and percentage. * Median and range values. ^‡^ Median and Interquartile range (IQR). ^¥^ Data referred just to women. CCIS: Cleveland Clinic Incontinence Score; RP: Resting pressure; MSP: Maximum squeeze pressure; SL: Sphincter length; LNR: Laboratory normal reference; EAS: External anal sphincter; IAS: Internal anal sphincter; OSph: Overlapping Sphincteroplasty; ALev: Anterior levatorplasty; TPFR: Total Pelvic Floor Repair.

**Table 2 jcm-11-03755-t002:** Comparison of preoperative values of CCIS with that of the different postoperative periods.

	CCIS					95%-CI			
Time	*n*	Preop.	SD	Postop.	SD	Mean Dif.	Inferior	Superior	*p*
12 m	117	17.9	2.3	2.8	3.3	15.1	14.4	15.9	<0.001
24 m	116	17.9	2.3	3	3.4	15	14.2	15.8	<0.001
36 m	107	18	2.1	2.6	3	15.4	14.6	16.2	<0.001
60 m	95	18	2.1	2.8	3	15.2	14.4	16	<0.001
120 m	60	18.1	2.1	2.4	1.9	15.8	15.1	16.5	<0.001
Last F-U	105	18	2.3	3.7	3.9	14.3	13.5	15.1	<0.001

CCIS: Cleveland Clinic Incontinence Score; 95%-CI: 95% Confidence Interval; Preop.: Preoperative; Postop: Postoperative; SD: Standard Deviation; Mean Dif.: Mean Difference; m: months; Last F-U: Last Follow-up.

**Table 3 jcm-11-03755-t003:** Univariate analysis of possible factors that could influence clinical results in terms of additional treatment necessity or global success at end follow-up time.

		Need of Additional Treatments			Satisfactory Results (>50% Reduction in CCIS)		
		No *n* (%)	Yes *n* (%)	*p*	No *n* (%)	Yes *n* (%)	*p*
Age (years) *		53.9 (17)	56.4 (13)	0.4	56 (14)	53.8 (15)	0.67
Sex	Female (113)	73 (64.6)	40 (35.4)	0.71	8 (7.6)	97 (92.4)	0.45
	Male (7)	5 (71.4)	2 (28.6)		0 (0)	7 (100)	
Duration of FI (months) *		55.4	39.6	0.29	51.1	49.2	0.95
Location of Sphincter Tear	Anterior (104)	67 (64.4)	37 (35.6)	0.49	7 (7.2)	90 (92.8)	0.71
	Posterior or Lateral (16)	11 (68.8)	5 (31.3)		1 (6.7)	14 (93.3)	
Tear Grades *		126 (27)	126 (30)	0.93	115 (30)	128 (27)	0.27
IAS Repair	Yes (84)	54 (64.3)	30 (35.7)	0.8	4 (5.1)	74 (94.9)	0.21
	No (36)	24 (66.7)	12 (33.3)		4 (11.8)	30 (88.2)	
Levatorplasty	Yes (82)	55 (67.1)	27 (32.9)	0.48	6 (7.8)	71 (92.2)	0.69
	No (38)	23 (60.5)	15 (39.5)		2 (5.7)	33 (94.3)	
Associated Surgical Techniques	Yes (33)	25 (75.8)	8 (24,2)	0.13	2 (6.7)	28 (93.3)	0.9
	No (87)	53 (60.9)	34 (39.1)		6 (7.3)	76 (92.7)	

* Figures represent mean values and standard deviation; *p* value is for Student’s *T* test. CCIS: Cleveland Clinic Incontinence Score; M: Male; F: Female; FI: Fecal Incontinence; IAS: Internal Anal Sphincter.

## Data Availability

Data supporting reported results are not hosted in any public data repository but will be shared if it is requested by any interested researcher.

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
