# Peer review of "Multimodal Management of Fecal Incontinence Focused on Sphincteroplasty: Long-Term Outcomes from a Single Center Case Series"

_jcm, 2022, doi:10.3390/jcm11133755_

Round 1
Reviewer 1 Report
This is an original article concerning the multimodal management of fecal incontinence
- The topic is promising and very much debated. Congratulations to the authors for their efforts and work.
- The introduction should be modified and focused on the main topic, i.e. sphincteroplasty
- Were the patients consecutive?
- The results are well described and developed and the figures have an excellent educational value
- Has the scar been removed? why was a monofilament (prolene) not used for the plasty? when the authors realized that dissection and muscle mobility were sufficient for a tensione-free repair?
Author Response
Reviewer #1
Comments and Suggestions for Authors
This is an original article concerning the multimodal management of fecal incontinence
- The topic is promising and very much debated. Congratulations to the authors for their efforts and work.
Thank you very much for the comment. We appreciate it, as it has been a long way to get to this point.
- The introduction should be modified and focused on the main topic, i.e. sphincteroplasty
Dear colleague. Thank you for the comment. We included some additional sentences and paragraphs to complete the introduction focusing on sphincteroplasty. However, as it can be considered that the great competitor to it is SNM, mainly based on deterioration of results of sphincteroplasty, with decided to keep that.
Firstly described by Parks and McPartlin [1], the traditional surgical management in patients presenting with fecal incontinence (FI) where there is an external anal sphincter (EAS) defect is an overlapping sphincteroplasty [2]. Although short- and medium-term functional outcomes after surgery are acceptable, some reports have shown that patients can anticipate a significant fall-off in the success of sphincter repair over a more prolonged follow-up [3-5]. Given the multifactorial nature of FI, the repair of a demonstrable sphincter defect remains valid [6]. There is considerable evidence that the success of sphincteroplasty, however, is dependent upon the level of experience of the surgeon [7] and the grade of the sphincter tear [8]. Nevertheless, although results with this technique appeared to deteriorate over time, the percentage of patients who remained satisfied was high [5].
- Were the patients consecutive?
Yes, all the included patients in the analysis were consecutive. We included that information in the methods section as follows.
All consecutive patients operated on for fecal incontinence who underwent a sphincter repair were included.
- The results are well described and developed and the figures have an excellent educational value
Thank you very much again for your positive comments.
- Has the scar been removed? why was a monofilament (prolene) not used for the plasty? when the authors realized that dissection and muscle mobility were sufficient for a tensione-free repair?
Thank you for these interesting “easy” but quite interesting and relevants questions. We have included some additional information to clarify all these issues, both in the material and methods and the discussion sections as considered appropriate in each case.
Material and methods:
- When there existed an important or pathologic scar, it was completely excised in order to get a free tension sphincter and to allow further healing over a healthy tissue.
- The EAS was minimally mobilized in order to avoid denervation and/or devascularization, but enough to allow a tension-free sphincteroplasty with an overlapping 2-layer sphincter suturing with 2/0 Vicryl sutures. During this part of the intervention sequential checking of tension is mandatory in order to identify that critical moment of not extensive but enough dissection of both EAS limbs.
Discussion:
- It has been authors practice to use absorbable sutures as we have always considered they produce less local tissue inflammation and granuloma, what might worsen healing and, eventually the final result. Apart from this consideration, strained or monofilament choice should not determine the final result
Reviewer 2 Report
Dear Authors,
Thank you for your interesting manuscript :
1. Can you please suggest the optimum time of the repair ?
2. Will a repair at the time of the injury will result in better long term outcomes ?
3. What is the optimum suture material for the repair?
4. Can you please state some relative and absolute contraindications for a sphincter repair based on your experience ?
Author Response
Reviewer #2
Dear Authors,
Thank you for your interesting manuscript :
- Can you please suggest the optimum time of the repair?
Thank you for the comment. We included this question to the discusión section.
Obviously, recognition of the sphincteric injury, whether due to obstetric trauma or any accident or violent insult of any kind, the sphincter must be repaired immediately. However, it is quite common that most patients with anal incontinence discovered later; in our series, almost half of the patients were older than 60 years. Therefore, sphincter repair should be performed based on incontinence severity, with no differences found regarding the time elapsed since the injury or patients´ age.
- Will a repair at the time of the injury will result in better long term outcomes?
Regarding the time to perform the sphincteroplasty, it is obvious that live recognition of the sphincteric injury, whether due to obstetric trauma, accident of any kind or violent insult, it should be repaired immediately, although there do not exist comparative data to answer this issue; whether an immediate or a delayed repair could result in better outcome has not previously been analyzed and therefore any judgement in this regard would be mostly lucubrating. Nevertheless, it is quite common that most patients with anal incontinence discovered later; in our series, almost half of the patients were older than 60 years, with a median duration of symptoms of 120 months. Therefore, sphincter repair should be performed based on incontinence severity, with no differences found regarding the time elapsed since the injury or patients´ age.
- What is the optimum suture material for the repair?
It has been authors´ practice to use absorbable sutures as we have always considered they produce less local tissue inflammation and granuloma, what might worsen healing and, eventually the final result. Apart from this consideration, strained or monofilament choice should not determine the final result.
- Can you please state some relative and absolute contraindications for a sphincter repair based on your experience?
Thank you for the question. It is difficult to talk about contraindications in this field, because there is not much evidence, but we will try to give a response based on our experience and the interpretation of literature.
In the event of anal incontinence, it is necessary to carry out an individualized assessment. If there is a sphincteric injury, and conservative measures have failed, surgical treatment is indicated in our group, following the previously mentioned systematics. However, some exceptions can be also being considered. As absolute contraindication a separation of the sphincteric margins > 180º or multiple sphincteric injuries can be mentioned. Regarding relative contraindications, it would be necessary to determine them personally in each patient, but severe defecatory dyssynergia or decreased rectal sensation verified by anorectal manometry can be prioritized for alternative treatments better than sphincteroplasty as first line. Nevertheless, in general terms, if there is a sphincteric injury, our opinion is that it is necessary to carry out the most complete and satisfactory anatomical repair possible and, based on the results, establish complementary treatments as necessary, as it has been previously exposed in our work.